# Web Retrieval Agents
# for Evidence-Based Misinformation Detection

**Jacob-Junqi Tian**
McGill | Mila

**Hao Yu**
McGill | Mila

**Yury Orlovskiy**
UC Berkeley

**Tyler Vergho**
Dartmouth

**Mauricio Rivera**
Mila

**Mayank Goel**
IIT Hyderabad

**Zachary Yang**
McGill | Mila

**Jean-François Godbout**
Université de Montréal | Mila

**Reihaneh Rabbany**
McGill | Mila

**Kellin Pelrine**
McGill | Mila
kellin.pelrine@mila.quebec

## Abstract

This paper develops an agent-based automated fact-checking approach for detecting misinformation. We demonstrate that combining a powerful LLM agent, which does not have access to the internet for searches, with an online web search agent yields better results than when each tool is used independently. Our approach is robust across multiple models, outperforming alternatives and increasing the macro F1 of misinformation detection by as much as 20 percent compared to LLMs without search. We also conduct extensive analyses on the sources our system leverages and their biases, decisions in the construction of the system like the search tool and the knowledge base, the type of evidence needed and its impact on the results, and other parts of the overall process. By combining strong performance with in-depth understanding, we hope to provide building blocks for future search-enabled misinformation mitigation systems.

## 1 Introduction

Misinformation and disinformation are significant societal challenges that threaten to intensify with progress in generative AI (Chen & Shu, 2023). Recent work (Chen & Shu, 2023; Pelrine et al., 2023) has shown that Large Language Models (LLMs) can effectively detect misinformation and provide a potential path to mitigate harm at scale. However, LLMs suffer from hallucinations and often lack knowledge of recent events due to a fixed training data window. These problems can be significantly mitigated if models have access to external sources of information. In this work, we propose a method to retrieve and leverage evidence from the web for misinformation detection.

Surprisingly, there are relatively few tools for Retrieval-Augmented Generation (RAG) that combine external data sources with recent LLMs in the domain of misinformation detection (Chen & Shu, 2023). In this study, we aim to build a strong, comprehensive approach that combines LLMs and RAG by integrating natural language understanding techniques for claim decomposition (Yao et al., 2022; Min et al., 2023; Chern et al., 2023; Zhang & Gao, 2023). In particular, we prompt an LLM to generate queries, and then answer them using another LLM connected to a web search engine. We evaluate the performance of this approach across a wide range of models: Vicuna, Mixtral, Claude, GPT-3.5, and two versions of GPT-4. We find that web-retrieval techniques improve the performance of all models except Vicuna; we also confirm that this improvement generally increases with the performance of the underlying model. In addition, we compare two search approaches—Cohere Coral with web connector, and DuckDuckGo with GPT-3.5 summarization—and confirm that they are both effective. Thus, our results indicate that the fully customizable DuckDuckGo tool can be used in future work, along with the more locked-in Cohere one; they also suggest that our

framework can be generalized to other search engines as well. We then analyze the sources retrieved, showing that: (1) having more sources is better; (2) the pipeline is not overly reliant on any one source; and (3) the sources it naturally uses exhibit little bias overall and high credibility. We continue investigating different components of the pipeline, including the summarizer and the knowledge base. We then turn to how and when web-searches are effective, showing that performance varies depending on the missing information and that some types of missing information may even make web searches counterproductive, which may point towards ways to improve both effectiveness and efficiency in RAG. We proceed to investigate uncertainty quantification, and how enabling web-search impacts the system's calibration capabilities, through a direct confidence elicitation prompt. Finally, we examine several datasets where web searches maintain but do not improve performance, showing its limitations and highlighting how search can be useful even without performance improvements.

Our key contributions are the following:

- We build a search framework that significantly improves performance for misinformation detection across multiple models, up to 20% depending on the model used (Table 2), outperforming other search frameworks.
- We conduct extensive analyses of different options within this framework, showing effective options (e.g., one can use either pre-built Cohere or customizable DuckDuckGo) and ones that should be avoided (e.g., using Wikipedia instead of the open web as knowledge base).
- We go beyond pure performance with in-depth analysis of sources used, biases, different types of missing evidence, and other aspects of the pipeline. This analysis brings the system closer to real-world viability and highlights limitations and opportunities for future work.

## 2 Related work

Many studies have shown the benefits of retrieving information to augment fact-checking and misinformation detection (Bekoulis et al., 2021; Kondamudi et al., 2023; Zhou & Zafarani, 2020). However, much of this past work integrates retrieval with older models like BERT (Liao et al., 2023), BART (Sundriyal et al., 2022), and memory networks (Ebadi et al., 2021; Ying et al., 2021). We note that these approaches have not demonstrated strong enough performance improvements to solve the problem of misinformation detection. Indeed, the task of detecting fake news or false information is challenging due to its ambiguous nature, where misinformation often contains a mix of true and false statements, making it difficult to discern the truth. It also requires up-to-date knowledge of real-world events and a nuanced understanding of deception techniques employed by malevolent actors who want to avoid detection. Nevertheless, LLMs have emerged as very promising research avenues for this task (Chen & Shu, 2023; Pelrine et al., 2023). Recent work has confirmed that these models still struggle with insufficient context, ambiguous inputs and hallucinations (Pelrine et al., 2023; Hsu et al., 2023; Orlovskiy et al., 2024), but identifying the missing information and providing additional context significantly improves the performance of these models on ambiguous statements (Orlovskiy et al., 2024; Pelrine et al., 2023). There is, therefore, a need to integrate strong retrieval tools into the new, LLM-based misinformation mitigation systems.

One particularly promising approach to LLM-based systems in the misinformation detection and fact-checking domains is decomposing inputs of uncertain veracity, such as statements or articles, into smaller units like claims. This enables chain-of-thought (Wei et al., 2022) reasoning particularly adapted to these tasks. For example, the ReAct framework (Yao et al., 2022) improves performance on the HotPotQA dataset (Yang et al., 2018) by decomposing its reasoning trace into a series of actions and observations traces for each thought generated (Yao et al., 2022). Likewise, the HiSS prompting method instructs the model to decompose a claim into several subclaims, that are then verified individually via multiple question-answering steps (Zhang & Gao, 2023). Three common approaches include 'FactScore', which decomposes claims into 'atomic facts' (Min et al., 2023), along with 'FacTool' (Chern et al.,

2023) and 'FOLK' (Wang & Shu, 2023), which are both designed to extract individual claims from longer statements.

Although several studies have highlighted the benefits of using RAG for LLMs in general applications (Saad-Falcon et al., 2023; Bendersky et al., 2023; Wang et al., 2024; Gao et al., 2024), there are surprisingly few analyses that apply this technique to misinformation detection and even fewer that combine it with decomposition. In a survey of over 650 works related to misinformation and LLMs (Chen & Shu, 2023), only two RAG misinformation detection methods were discussed (Chern et al., 2023; Cheung & Lam, 2023). The first focused on factuality in more general problems like mathematics and code generation, while the latter did not use decomposition. Given that decomposition and RAG could provide both better reasoning and evidence, we believe that such an approach will greatly improve the performance of LLMs for misinformation detection.

The closest work to this approach is HiSS (Zhang & Gao, 2023). HiSS prompts the LLM to decompose the claim with a few-shot demonstration, and triggers search if the LLM is not confident to answer directly. It outperforms other systems like ReACT (Yao et al., 2022) in this domain. However, we found that its effectiveness is inconsistent, particularly as its performance gets worse with better models (GPT-3.5 vs. GPT-4; see Section 4). Thus, in this study, we aim to develop a stable framework that performs well across a range of models. We also seek a deeper understanding of the search process, including when it helps, what sources get used, and how different design choices affect it.

## 3   Methodology

We enable information retrieval by explicitly permitting the main LLM agent to invoke search multiple times and encouraging the LLM to perform extensive reasoning before giving a prediction. This leads the LLM to naturally produce an effective decomposition of the task, in contrast to pipelines like FactLlama (Cheung & Lam, 2023), which invoke search only on the input. We perform the search with another LLM agent with a web connection and return the search agent's output to the main agent. The main agent repeatedly triggers this process until it decides to give a final prediction (for an example of a full input and output, see Appendix A.11). We test this pipeline across several recent models (GPT-4, GPT-3.5, Cohere Coral, Mixtral, Claude 3, and Vicuna-1.5), focusing particularly on the LIAR-New dataset (Pelrine et al., 2023).

### 3.1   Enabling search

Here we describe how we enable instruction-tuned generative LLMs to "search" online, especially those that do not explicitly support function calling. In the prompt, we inform the model that it has access to a search tool, and provide instructions on how to invoke it:

```
You have access to a search engine tool. To invoke search,
    begin your query with the phrase "SEARCH: ". You may
    invoke the search tool as many times as needed.
```

We parse the generator LLM output to extract the prescribed pattern. While invoking the search function, we supply the text that comes right after "SEARCH: " to the search pipeline. That pipeline does not have access to the original statement that we want to verify.

We explored two different implementations of the search pipeline: with the Cohere "Chat with RAG" API, and by summarizing the output of the DuckDuckGo search API.

In the Cohere pipeline, we invoke the Cohere Chat API with the official "web-search" connector enabled. We ask it to "Look up info" with the following code and template:

```
response = cohere.Client.chat(
    model="command",
    message="Look up info about the following: {query}",
    connectors=[{"id": "web-search"}]
)
```

Above, the placeholder "{query}" in the message to Cohere "Chat with RAG" would be replaced with the text extracted from the generator LLM output. The output of this search agent is a summary of the search results in natural language. We prepend the output with "Search result: " and add it to the conversation context as a new message with role user, then resume the main (offline) agent's generation. This results in interactions between the main agent and search agent as in Table 1.

| Component | Description |
|---|---|
| Generator | Analysis: Let's first search for the historical price of oil in June 2008 and March 2022. SEARCH: Historical price of oil in June 2008 |
| Query | Historical price of oil in June 2008 |
| Cohere Response | Oil peaked at $126.33 (inflation-adjusted price of $177.26) in June 2008. The price of Brent crude was $132.32 per barrel. This was the absolute peak, from which the price dropped sharply to $31.04 in February 2009. This was one of the sharpest drops in oil history due to the financial crisis, which caused energy prices to fall due to diminishing demand, contraction of credit with which to make purchases, and lower corporate earnings which led to layoffs and increased unemployment. |
| Summarized DuckDuckGo Response | Search result: The search results provide information on the price of oil in June 2008. CNN Business reported that oil prices settled above $140 a barrel for the first time on June 27, 2008, during a thinly traded session. This was influenced by a selloff on Wall Street that sent the Dow into bear market territory . . . The specific price of Brent crude in June 2008 is mentioned in one search result as $132.32 per barrel. Overall, the search results provide diverse information about the price of oil in June 2008. However, not all results mention the specific price for that month, and some focus more on the broader context and impact of the 2008 financial crisis on oil prices. |

Table 1: Example of parsed search query and response from Cohere "Chat with RAG" API.

**DuckDuckGo**  As an alternative to the Cohere service, we utilize the DuckDuckGo search engine through its web search API.[1] We chose this search engine because there is a free API, unlike, for example, Google and Bing. We gather the titles and content of the top 10 search results of the query and use GPT-3.5 Turbo to summarize the content with the following prompt.

```
Please summarize the searched information for the query. Summarize
    your findings, taking into account the diversity and accuracy
    of the search results. Ensure your analysis is thorough and
    well-organized.
Query: {query}
Search results: {results}
```

**Wikipedia retrieval with local embeddings**  We also implemented a local retrieval system containing all Wikipedia English pages (cutoff at 2023.11) with the *bge-large-en-v1.5* embedding model. The embeddings of all chunks are indexed with the FAISS (Douze et al., 2024) library. The queries are generated with the same offline agent setup, and the retrieved chunks are passed through the summarization system and then back to the offline agent, as described above. This dense retrieval system provides an alternative to DuckDuckGo and similar search engines. In particular, it provides a comparison of open web search, which is increasingly popular and enabled by recent LLMs, with more traditional fixed knowledge base retrieval mechanisms, using one of the most widely leveraged knowledge bases for such systems.

---

[1]Python Package: `duckduckgo_search`

## 3.2 Extracting predictions

We prompt the main agent to return a binary number as the prediction:

```
... state "True statement; Factuality: 1" if you think the
    statement is factual, or "False statement; Factuality: 0"
    otherwise.
```

We repeat each experiment 5 times and report the 95% confidence interval over the five F1 scores (this generally gives a larger but more thorough interval than the commonly reported standard error).

## 3.3 Baselines

**Offline models**   We compare the baseline "offline" performance of the models with our own search enabled framework. The prompt for the offline version is as follow:

```
Your task is to analyze the factuality of the given statement.
    After providing all your analysis steps, summarize your
    analysis and state "True statement; Factuality: 1" if you
    think the statement is factual, or "False statement;
    Factuality: 0" otherwise. You should begin your summary with
    the phrase "Summary: ". Statement: {statement}
```

**Cohere Chat with RAG**   Note that in the main experiment setup, we do not provide the Cohere "Chat with RAG" model access to the original statement that we seek to verify. Similarly, for DuckDuckGo, we also just prompt the modified query, instead of the original statement. In both cases, this ensures we are leveraging our decomposition framework. However, the Cohere model which has been set up for web search could potentially do it effectively on its own, without the help of the generator LLM, if it were given the original statement. Hence, we implement it as a baseline to determine if the "search" pipeline adds any benefit beyond just invoking Cohere Chat with RAG directly. To do this, we query Cohere "Chat with RAG" with the same prompt given to other generator LLMs when the search action is disabled (see the above prompt).

**HiSS**   We selected this baseline as it provided the state-of-the-art decomposition and web search approach in this domain. We implemented HiSS (Zhang & Gao, 2023) using the updated code provided by the authors. We test both the original version which predicted 6-way labels (also converted to binary to match our other evaluation) and a direct binary version where we minimally changed the prompt by replacing the part that lists the 6 labels with "true and false". As the underlying model, we use GPT-3.5 like the original paper, and we also test GPT-4. We use the most recent version of both at the time of this writing, 0125.

**WikiChat**   We implement WikiChat (Semnani et al., 2023), a model grounded on the English Wikipedia that outperforms other models on factual accuracy retrieval baselines, using the code provided in the original repository.[2] However, instead of using text-davinci-003 as the engine model like in the original paper, we use GPT-3.5-turbo-instruct because text-davinci-003 was deprecated by OpenAI. We follow the configuration suggested in their code repository to obtain results comparable to the GPT-3.5 results reported in their original paper. The prompt used for WikiChat is as follows:

```
Your task is to analyze the factuality of the given statement.
    After providing all your analysis steps, summarize your
    analysis and state "True statement; Factuality: 1" if you
    think the statement is factual, or "False statement;
    Factuality: 0" otherwise. You should begin your summary with
```

---

[2]https://github.com/stanford-oval/WikiChat

```
the phrase "Summary: " and conclude your response with "
Factuality: 1" or "Factuality: 0". Statement: {statement}
```

This matches the original except we explicitly specify concluding with "Factuality: 1" or "Factuality: 0", because we noticed that without this, the refining step would strip the score from the end of the model's response and lead to parsing errors.

## 3.4 Uncertainty quantification

We investigated the system's uncertainty quantification performance through a direct confidence elicitation methodology (Lin et al., 2022), which has been previously demonstrated to be effective in the misinformation detection context (Pelrine et al., 2023; Rivera et al., 2024). Specifically, we prompted the generative LLMs to provide an uncertainty estimation of its analysis once we extracted the system's analysis and predictions:

```
Statement: {misinformation statement}
Proposed analysis: {web-retrieval agent's analysis}
Your task is to rate the uncertainty of the proposed analysis on a
    score from 0 to 100, where 0 represents definitely
    uncertainty and 100 represents definitely certain. Please,
    only answer with your score.
```

## 3.5 Datasets

Our main dataset is LIAR-New (Pelrine et al., 2023). This dataset consists of statements from the fact-checking website PolitiFact.org. It was chosen because in addition to statements and factuality labels, it also contains labels on examples that are missing context. In particular, statements are classified as Possible, Hard, or Impossible based on whether it should be possible to evaluate the factuality without additional input context.

While the raw LIAR-New dataset provides six factuality categories, we follow common practice (Pelrine et al., 2023) and map the "half-true", "mostly-true", as well as "true" levels as 1 ("True"). The other three factuality categories – "false", "barely-true", and "pants-fire" – are labelled as 0 ("False"). Given that the resulting dataset is skewed towards label 0, we report macro F1 instead of accuracy for each experiment. We randomly sample 588 examples from the larger dataset to reduce compute and API costs.[3] The label distribution of the sample, before binarizing, is given in Appendix A.1.

Besides LIAR-New, we also test performance on FEVER-v2 (Thorne et al., 2019), which is a fact-checking dataset based on Wikipedia claims but constructed to be more challenging and adversarial than the original FEVER dataset (Thorne et al., 2018). On this dataset we compare with the recent state-of-the-art method of Zhang et al. (2024) which uses a multi-hop approach with a claim-evidence graph. Additionally, we also evaluate performance on the FEVER, FaVIQ (Park et al., 2022) and X-FACT (Gupta & Srikumar, 2021) fact-checking datasets. The latter is multilingual.

## 4 Results

**Performance of search** We see in Table 2 that when web searches are enabled, `gpt-3.5-turbo`, `gpt-4-0613`, `gpt-4-0125`, and `claude3-haiku` all surpass the performance of the Cohere Chat baseline on the LIAR-New dataset. They likewise surpass the performance of WikiChat (Semnani et al., 2023) and HiSS (Zhang & Gao, 2023), the latter of which becomes worse with GPT-4 instead of GPT-3.5 (see also Appendix A.2 showing it is inconsistent in returning parsable output). Performance of `mixtral` also improves substantially

---

[3]Given the additional costs related to the non-turbo `gpt-4-0613` (knowledge-cutoff 2021/03), we further sub-sample the dataset to keep only 100 examples in the two experiments involving this model, as well as with HiSS GPT-4.

| Baseline Model | Search |
|---|---|
| Cohere Chat with RAG (3.3) | 63.9% ± 3.5% |
| HiSS GPT-3.5 (Zhang & Gao, 2023) | 60.6% |
| HiSS GPT-3.5 Binary (Zhang & Gao, 2023) | 62.7% |
| HiSS GPT-4 (Zhang & Gao, 2023) | 56.1% |
| WikiChat GPT-3.5 (Semnani et al., 2023) | 54.0% |

| Our Model (Knowledge Cutoff) | No Search | Search via Cohere RAG (↑) | ΔF1 |
|---|---|---|---|
| vicuna-13b-v1.5 | 58.4% ± 6.4% | 57.5% ± 2.1% | −0.9% |
| mixtral-8x7b-instruct | 52.9% ± 7.6% | 58.6% ± 7.6% | +5.7% |
| gpt-3.5-turbo (2021/03) | 59.3% ± 5.8% | 64.7% ± 5.3% | +5.4% |
| gpt-4-0613 (2021/03) | 47.8% ± 9.2% | 68.3% ± 14.5% | **+20.5%** |
| gpt-4-0125 (2023/12) | 58.9% ± 7.7% | **71.7% ± 4.5%** | +12.8% |
| claude3-haiku (2023/08) | **64.1% ± 3.6%** | 71.3% ± 6.6% | +7.2% |

Table 2: F1 score of models with and without search. Search performs substantially better.

with our method. `claude3-haiku` shows the best performance without search, and is on-par with GPT-4 with search. Additionally, in Appendix A.8, we check sensitivity to some slight prompt variations, and find that search consistently improves performance.

In Table 3, we see that the DuckDuckGo search pipeline also substantially improves the model's performance. On average it is slightly worse than Cohere, but the results are close, especially with `gpt-4-0125`.

In Appendix A.3, we examine other choices of summarizer in the DuckDuckGo pipeline. We find that a stronger model for the summarizer improves overall performance.

| Model Name (Knowledge Cutoff) | No Search | Summarized DuckDuckGo Search (↑) | ΔF1 |
|---|---|---|---|
| mixtral-8x7b-instruct | 52.9% ± 7.6% | 56.9% ± 3.1% | +4.0% |
| gpt-3.5-turbo (2021/03) | 59.3% ± 5.8% | 60.3% ± 9.9% | +1.0% |
| gpt-4-0125 (2023/12) | 58.9% ± 7.7% | **70.3% ± 8.5%** | **+11.4%** |
| claude3-haiku (2023/08) | 64.1% ± 3.6% | 67.1% ± 6.6% | +3.0% |

Table 3: F1 score of models with and without DuckDuckGo search + GPT-3.5 summary. Search again improves performance substantially.

| Main Agent Model Name | Duck +PF (k=10) | Duck −PF (k=10) | Duck −PF (k=5) | Duck −PF (k=2) |
|---|---|---|---|---|
| gpt-3.5-turbo (2021/03) | 60.3% ± 9.9% | 60.3% ± 4.9% | 58.7% ± 4.7% | 56.5% ± 4.0% |
| gpt-4-0125 (2023/12) | 70.3% ± 8.5% | 66.9% ± 6.5% | 64.7% ± 4.3% | 63.7% ± 3.4% |

Table 4: F1 scores on LIAR-New (subsampled to one third to reduce cost) with PolitiFact allowed (+PF) or disallowed (−PF) as a source. We vary numbers of DuckDuckGo search results (k=10,5,2). The pipeline still performs well without PolitiFact. More search results increase performance.

**Sources** LIAR-New is collected from PolitiFact, raising a natural question about how much role PolitiFact plays when doing web search. In real-world applications, we would both like the system to take advantage of PolitiFact when it is available, and still do well when it is not (e.g., for new misinformation that PolitiFact has not yet fact-checked). In Appendix A.5.1 we find that PolitiFact is the most frequently used source, searched nearly as much as the rest of the top 10 sources combined.

To simulate PolitiFact not being available, in Table 4, we test blocking this website and relying on other sources. This is done with the DuckDuckGo pipeline, where we have more control than the Cohere one. We find that blocking PolitiFact does not degrade performance of GPT-3.5, and while performance of GPT-4 does drop a bit, it remains significantly better than without search. We also perform an ablation on the number of results we set the search to return, and find performance decreases monotonically with fewer results. Putting these results together, along with the fact that PolitiFact is the ideal source to provide evidence for this dataset, these findings suggest that no single source is necessary for accuracy, but it is important to have enough sources to reduce noise and improve the chances of finding the necessary evidence.

In Appendix A.5.2, we study the bias of the sources and input statements. We find that the sources used are slightly but not substantively left leaning (-0.54 on a scale from -3 to 3, putting it between 1="center-left" and 0="center"), while the input statements are right leaning to an almost exactly equal degree (0.53 on the same scale). Thus, our system does not exhibit substantial bias in terms of sources used. To assess the overall quality of the sources, we evaluate the websites accessed by the web retrieval system using the comprehensive media source evaluation database (Lin et al., 2023), noting that the average quality of the sources is in the range of 0.76-0.80 for all models, roughly on par with the New York Times (0.87). In the same Appendix, we examine the bias and overall domain quality versus performance of our search systems, but do not find conclusive results there.

| Wiki Full Chunk k=10 | WikiLocal k=10 | WikiLocal k=20 | WikiChat GPT-3.5 (2023) | GPT-3.5 Duck −PF |
|---|---|---|---|---|
| 50.1% ± 11.0% | 49.9% ± 8.5% | 48.9% ± 3.4% | 54.0% | 60.3% ± 4.9% |

Table 5: F1 score of models using local embedding-based retrieval from Wikipedia vs. web search. We see these approaches are substantially worse than web search ones, suggesting Wikipedia is not an appropriate knowledge base for these types of data.

**Knowledge base**   Table 5 provides additional analysis on Wikipedia as knowledge base vs. the open web. Specifically, we compare the performance of using Wikipedia retrieval systems with our offline GPT-3.5 agent (specifically, full chunks retrieval with k=10 and chunked articles retrieval with k=10 and k=20), the WikiChat system from the literature (Semnani et al., 2023), and our GPT-3.5 + DuckDuckGo search system. The chunk split uses an embedding collection for Wikipedia published on HuggingFace.[4] We see that all the Wikipedia-based systems – both in our setups and in a leading one from the literature (WikiChat) – provide inferior performance compared to our open web search systems. These results confirm the value of open web search and the limitations of a fixed knowledge base in this domain.

**Types of missing evidence**   We expect web retrieval to improve the factuality evaluation performance on statements that are missing a key piece of context. We leverage results from Orlovskiy et al. (2024) that classified Hard and Impossible statements (missing context) in this dataset by categories of missing information. We use these category labels to evaluate if the type of missing information in a statement affects web retrieval performance. Our approach evaluates performance for the following categories of missing information: speaker, location, textual evidence, non-textual evidence (photo, video, audio), and date. We calculate the difference between the F1 scores with and without web search for identical models, and analyze the improvement in performance by category. For this analysis where the data is split into bins, we determine the F1 scores based on the model's majority prediction across five runs for each statement, reducing variance.

Across the 4 generator models (`vicuna-13b-v1.5`, `mixtral-8x7b-instruct-v0.1`, `gpt-3.5-turbo`, `gpt-4-0125-turbo`) we see the highest average improvement for textual evidence, where retrieval is helpful across all models. We also note some improvement for missing non-textual evidence. This suggests retrieval can add some indirect multimodal capabilities to text-only systems, but it still performs better with text. There are also some

---

[4]Cohere/wikipedia-2023-11-embed-multilingual-v3

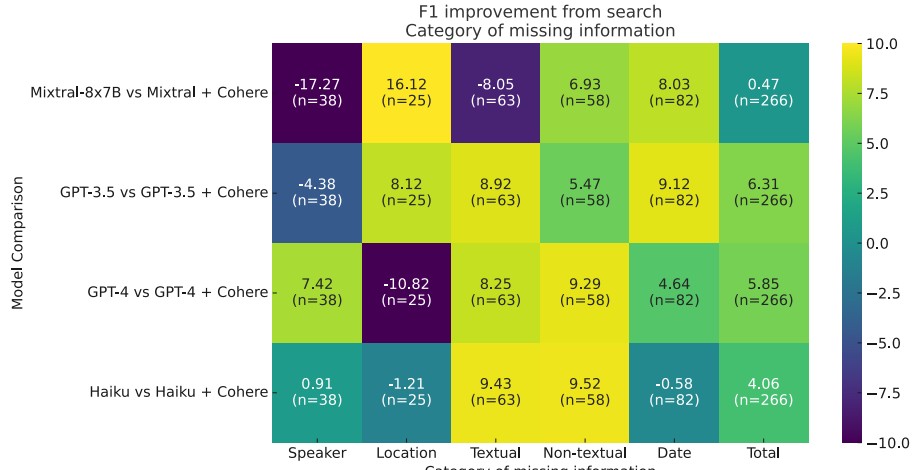

Figure 1: F1 improvement from Cohere RAG Search by category of missing information for each model, for statements labeled as hard and impossible to evaluate. The columns represent the category of missing information in each statement.

improvements in the date category. Other categories are more mixed, with some substantial drops in performance. We also see no difference in F1 on statements that are labeled as having sufficient context (p) in LIAR-New (Appendix A.4). This result suggests that web retrieval does not improve performance when a statement already has the necessary context to be evaluated for factuality.

We see particularly sharp drops in performance for some models with speaker and especially location categories. We hypothesize that this is due to finding the wrong information; for instance, the dataset is US-centric, and it has previously been observed that GPT-4 may assume US context when detecting misinformation (Pelrine et al., 2023). Therefore, we suspect that the performance drops when models retrieve incorrect context and stop assuming a US one. Speaker and location information might also simply be more difficult to retrieve without direct specification, because they create a variety of scenarios a statement may refer to. In Appendix A.4, we also further break these results down by comparing across Possibility categories and comparing Cohere vs. DuckDuckGo. Based on all these analyses, if we can selectively invoke web retrieval in the categories it is most effective with, we might both improve the overall performance and reduce computational costs. We plan to examine these possibilities in future work.

**Uncertainty quantification**    Given perfect performance remains unachievable, it is valuable to have an estimate of the uncertainty in a system's predictions (Pelrine et al., 2023). One would hope that enabling web-search would enhance the stability and certainty of those predictions. In Table 6, we compare the system's calibration performance with and without web-search through our confidence elicitation methodology. As hypothesized, for both gpt-3.5-turbo and gpt-4-0125 models, enabling web-search improves the system's uncertainty quantification performance. See Appendix A.6 for a visualization of the effect of web-search on the uncertainty scores. We also attempted to perform uncertainty quantification on the search results themselves, to create an estimator for whether they gave high quality and comprehensive information, but only found negative results (Appendix A.7).

**Limitations of search by data and task**    As seen in Table 7, there is no performance gain from search on the FEVER-v2 dataset. We hypothesize that the limited performance gain is largely because the facts in this dataset are much simpler and constrained in scope, comprising solely information from Wikipedia, rather than real-world misinformation narratives. Thus, LIAR-New requires looking up sources from a range of news websites. In comparison, FEVER-v2 is constructed entirely around articles on Wikipedia—a single,

| Model Name (with or without search) | ECE Score | Brier Score |
|---|---|---|
| gpt-3.5-turbo (no web-search) | $0.1600 \pm 0.0057$ | $0.1557 \pm 0.002$ |
| gpt-3.5-turbo (web-search enabled) | $0.1093 \pm 0.0135$ | $0.1322 \pm 0.001$ |
| gpt-4-0125 (no web-search) | $0.09 \pm 0.0057$ | $0.1237 \pm 0.0001$ |
| gpt-4-0125 (web-search enabled) | $0.0646 \pm 0.0035$ | $0.1113 \pm 0.0007$ |

Table 6: Effect of enabling web-search on uncertainty quantification of GPT models on LIAR-New, quantified by expected calibration error (ECE) and Brier score (in both cases, lower is better).

| Baseline Model | | Search | |
|---|---|---|---|
| Causal Walk (Zhang et al., 2024) | | 62.1 | |
| Cohere Chat with RAG (3.3) | | $72.0\% \pm 2.0\%$ | |
| **Our Model** (Knowledge Cutoff) | **No Search** | **Search via Cohere RAG (↑)** | **ΔF1** |
| gpt-3.5-turbo (2021/03) | $75.9\% \pm 2.0\%$ | $71.0\% \pm 1.2\%$ | $-4.9\%$ |
| gpt-4-0125 (2023/12) | **84.7**% | 84.6 | $-0.1\%$ |

Table 7: F1 score of models with and without search on FEVER-v2.

well-known website that is very likely to be part of the training set of the models we tested. We note that all the LLM-based approaches, search or not, greatly outperform recent state-of-the-art approaches like (Zhang et al., 2024). We also find similar results on FEVER (v1), FaVIQ and X-FACT (Appendix A.10). Thus, our search system is still valuable in contexts like these because it enables one to integrate the best performing model with verifiable evidence and citations. At the same time, we also find substantial variation in the average number of searches performed per example over the different datasets (Appendix A.9), with the system using less on other datasets than on LIAR-New, which might be a fruitful area for future research.

**Limitations of understanding by model**  Pairing the "Chat with RAG" model from Cohere with a more powerful main agent enabled us to exceed the performance of both Chat with RAG and the main LLM agent itself. We hypothesize that compared to gpt-3.5-turbo or the model behind Cohere's Chat with RAG (Command R, in our experiments), the gpt-4-turbo-0125 LLM is capable of doing a better job at decomposing the statement and generating queries. By asking better questions, the LLM retrieved information that was more relevant, which led to a better overall performance. As the knowledge cutoff of gpt-4-turbo-0125 (2023/12) encompassed the entirety of LIAR-New, we initially hypothesized that this main agent LLM would not benefit as much from search as gpt-3.5-turbo (2021/03), which knew less about the news events in LIAR-New. The results proved the opposite, suggesting again that strong queries are essential. Refer to Appendix A.11 for a concrete example and additional discussions. In future work, we plan to test the effect of decomposition and query generation vs final evaluation by taking the decomposition and search results of GPT-4 and making a final prediction with weaker models.

## 5  Conclusion

We have described a framework for leveraging web retrieval to detect misinformation that substantially improves performance. We have also confirmed its flexibility and customizability by showing that it works with multiple models and search tools. Finally, we have explored numerous parts of the process, including the sources and their biases, how different levels of search and summarizing affect the results, the impact of open web vs. restricted knowledge base, when search is effective and when it is not, and more. Taken together, we hope this line of work will lead to a practical understanding of how to build better, more evidence-based misinformation mitigation tools.

## 6    Acknowledgements

This work was partially funded by the CIFAR AI Chairs Program and the Centre for the Study of Democratic Citizenship. We also thank Berkeley SPAR for connecting collaborators and funding support. Kellin Pelrine was supported by funding from IVADO and by the Fonds de recherche du Quebéc.

## 7    Author Contributions

Jacob Tian proposed the initial two-agent architecture, built the core pipeline, and performed numerous experiments and analyses. He also took a leading role in the team to execute the project, and wrote large portions of the paper. Hao Yu set up and tested DuckDuckGo, blocking PolitiFact, different summarizers, and made a number of other substantial contributions to the experiments, writing, and analysis. Yury Orlovskiy performed extensive analysis of the sources and the effects of different missing information, and led the writing of those sections. Tyler Vergho ran the HiSS and WikiChat baselines, and contributed to the writing. Mauricio Rivera developed the uncertainty quantification method for the overall results, and compared the performance with and without search. Mayank Goel performed experiments on uncertainty quantification of search itself. Zachary Yang contributed suggestions, feedback, and helped write the paper. Jean-François Godbout and Reihaneh Rabbany advised the project, contributing wide-ranging ideas and feedback. Kellin Pelrine supervised the project, providing guidance and feedback across all stages and all components of the project.

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

# A    Appendix

## A.1    Label distribution

The label distribution is shown in Table 8.

| Mapped Label | Original Label | Count |
|---|---|---|
| False (0) | "Pants-Fire" | 103 |
| False (0) | "False" | 323 |
| False (0) | "Barely True" | 69 |
| True (1) | "Half-True" | 49 |
| True (1) | "Mostly-True" | 26 |
| True (1) | "True" | 18 |

Table 8: LIAR-New Label Distribution.

## A.2 Parsability of returned outputs

In Table 9, we reprint the main results of Table 2, with the addition of percentage (%) parsed which reflects how many of the outputs contain the necessary information to parse them and compare with the ground truth. Most models have a high parsability rate, which is slightly improved by adding the search option. Previous research (Pelrine et al., 2023) has observed LLMs like GPT-4 occasionally refusing to answer when it is not confident, so we hypothesize that the evidence found with search makes the models more confident about answering.

Some versions of HiSS, notably the GPT-4 one, suffer from a lower parsability rate.

| Baseline Model (Knowledge Cutoff) | | | Search | %parsed | |
|---|---|---|---|---|---|
| Cohere Chat with RAG (3.3) | | | $63.9\% \pm 3.5\%$ | *99.7%* | |
| HiSS GPT-3.5 (Zhang & Gao, 2023) | | | *60.6%* | *91.3%* | |
| HiSS GPT-3.5 Binary (Zhang & Gao, 2023) | | | *62.7%* | *99.2%* | |
| HiSS GPT-4 (Zhang & Gao, 2023) | | | *56.1%* | *91.0%* | |
| WikiChat GPT-3.5 (Semnani et al., 2023) | | | *54.0%* | *95.1%* | |

| Our Model (Knowledge Cutoff) | No Search | %parsed | Search via Cohere RAG (↑) | %parsed | ΔF1 |
|---|---|---|---|---|---|
| vicuna-13b-v1.5 | $58.4\% \pm 6.4\%$ | *99.8%* | $57.5\% \pm 2.1\%$ | *95.5%* | $-0.9\%$ |
| mixtral-8x7b-instruct | $52.9\% \pm 7.6\%$ | *99.1%* | $58.6\% \pm 7.6\%$ | *99.4%* | $+5.7\%$ |
| gpt-3.5-turbo (2021/03) | $59.3\% \pm 5.8\%$ | *96.7%* | $64.7\% \pm 5.3\%$ | *99.4%* | $+5.4\%$ |
| gpt-4-0613 (2021/03) | $47.8\% \pm 9.2\%$ | *90.2%* | $68.3\% \pm 14.5\%$ | *98.4%* | $+\mathbf{20.5\%}$ |
| gpt-4-0125 (2023/12) | $58.9\% \pm 7.7\%$ | *98.8%* | $\mathbf{71.7\%} \pm \mathbf{4.5\%}$ | *98.6%* | $+12.8\%$ |

Table 9: F1 score of models with and without search, including % parsed – i.e., how many of the outputs can be parsed for comparison with the ground truth.

## A.3 Summarizer

In Table 10 we perform an ablation on the summarizing model, comparing our default summarizer with gpt-3.5-turbo (GPT-3.5) against gpt-4-0125 (GPT-4) and mixtral-8x7b-instruct (Mixtral). We see that GPT-4 outperforms GPT-3.5 which outperforms Mixtral, suggesting that the more powerful the model used in the summarizer, the better the result. With GPT-4, DuckDuckGo with Politifact blocked performs on par with Cohere with no restrictions. Potentially, stronger summarizers are better at reconciling conflicting evidence – this might be an interesting topic for future research.

| No Search Action | Cohere Search | Duck −PF GPT-3.5 | Duck −PF GPT-4 | Duck −PF Mixtral |
|---|---|---|---|---|
| $59.9\% \pm 2.6\%$ | $65.1\% \pm 4.4\%$ | $63.5\% \pm 1.5\%$ | $64.8\% \pm 9.0\%$ | $62.8\% \pm 3.1\%$ |

Table 10: F1 score on LIAR-New with GPT-3.5 main agent, varying the summarization model after DuckDuckGo retrieval. More powerful summarization improves performance.

## A.4 Additional F1 difference heatmaps

In the following figures, we show additional heatmaps illustrating differences in performance for the search-enabled system versus the offline model. We omit categories with less than 20 samples.

For example, in figures 2a and 2b, we present F1 differences in gpt-4-0125 with and without search. We also break the results down by Possibility label. While often similar, we find some discrepancies. For example, with Cohere versus DuckDuckGo search for gpt-4-0125, there is a substantial difference in the location and date categories, where DuckDuckGo performs poorly. We suspect Cohere might also be US-centric, and DuckDuckGo less so, which would explain the location difference. We do not yet have a clear explanation for the difference in the date category. We believe further analysis of this type could help optimize different search engine queries, or even choose the optimal search engine for each type of query.

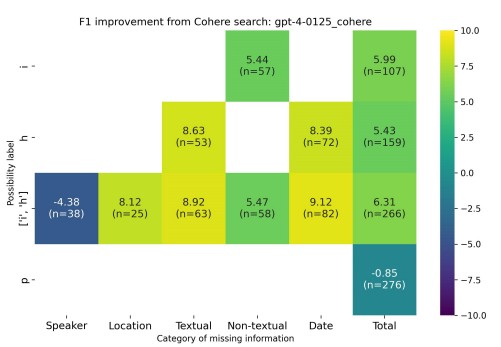

(a) F1 improvement with Cohere.

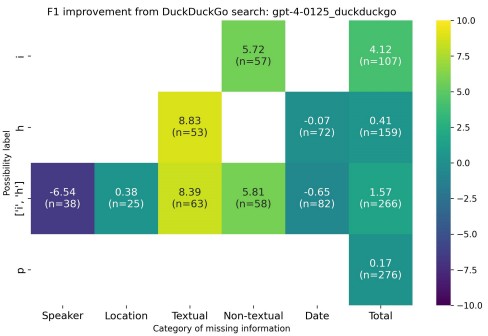

(b) F1 improvement with DuckDuckGo.

Figure 2: Improvement with Cohere vs. DuckDuckGo: gpt-4-turbo-0125. The rows correspond to Possibility label (i = Impossible, h = Hard, p = Possible).

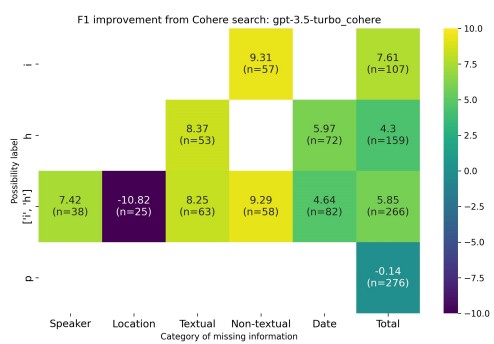

(a) F1 improvement with Cohere: gpt-3.5-turbo.

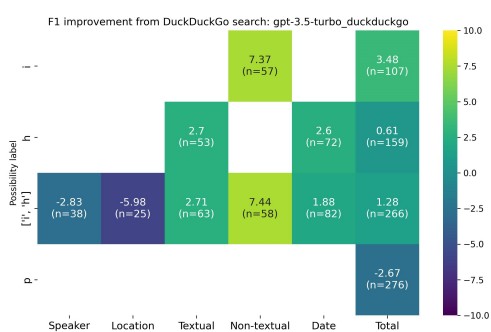

(b) F1 improvement with DuckDuckGo: gpt-3.5-turbo.

Figure 3: Improvement with Cohere vs. DuckDuckGo: gpt-3.5-turbo.

## A.5 Additional source and performance analysis

### A.5.1 Source frequency analysis

We analyze the most frequently accessed sources by the web retrieval agent when using the Cohere search system on the LIAR-New dataset. First, in Table 11, we look at the top 10 most used sources by GPT-3.5 and GPT-4. We note the counts are lower for GPT-4 compared

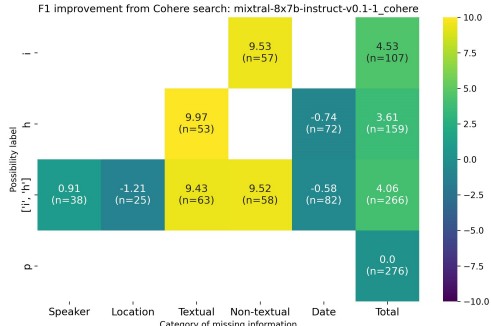

(a) F1 improvement with Cohere: mixtral-8x7b-instruct-v0.1-1.

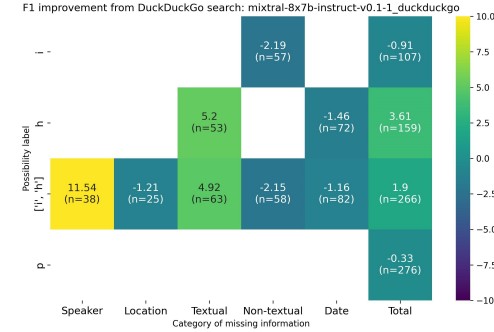

(b) F1 improvement with DuckDuckGo: mixtral-8x7b-instruct-v0.1-1.

Figure 4: Improvement with Cohere vs. DuckDuckGo: `mixtral-8x7b-instruct-v0.1-1`.

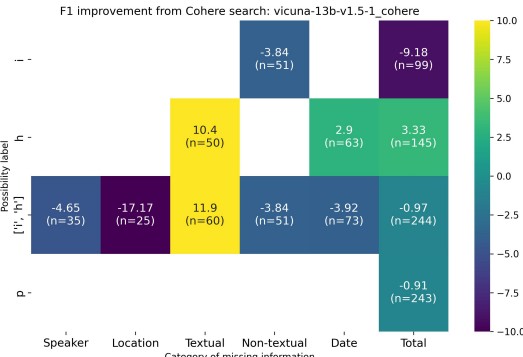

Figure 5: F1 improvement: vicuna-13b-v1.5-1.

to GPT-3.5, as GPT-3.5 was run on the entire dataset, while GPT-4 was run on a random third of this content to reduce costs. We see though that the sources used are very similar. PolitiFact dominates, followed by Wikipedia, and the rest of the top 10 comprise a mix of news and fact-checking websites. There are slight differences in the order, for instance, for GPT-3.5 CNN comes in at number 8, while for GPT-4 it is number 11 (not shown in table). But overall this confirms that the differences between the two systems in terms of which sources they examine are very small.

| GPT-3.5 | Count | GPT-4 | Count |
|---|---|---|---|
| politifact.com | 10475 | politifact.com | 3690 |
| en.wikipedia.org | 2924 | en.wikipedia.org | 648 |
| usatoday.com | 1321 | reuters.com | 477 |
| reuters.com | 1204 | usatoday.com | 340 |
| statesman.com | 1171 | apnews.com | 318 |
| apnews.com | 1116 | statesman.com | 280 |
| snopes.com | 806 | nytimes.com | 262 |
| cnn.com | 741 | snopes.com | 199 |
| nytimes.com | 718 | checkyourfact.com | 198 |
| checkyourfact.com | 607 | washingtonpost.com | 155 |

Table 11: Top 10 sources used by GPT-3.5 and GPT-4 with Cohere Search.

### A.5.2 Bias, credibility, and factuality analysis

We next go deeper on the quality and ideology of the sources used, as well as the input statements. First, we evaluate sources using Media Bias Fact Check (MBFC),[5] a non-partisan media assessment organization, which labels news websites according to their ideological biases. We used an API to retrieve the most current labels.[6] The bias ratings are as follows:

```
Bias:
-3 indicates an extreme left-wing favoring
-2 indicates a left favoring
-1 indicates a center-left wing favoring
0 indicates center favoring
1 indicates a center-right wing favoring
2 indicates a right favoring
3 indicates an extreme right-wing favoring
```

Analyzing the results with GPT-3.5, we find that 78% of the sources used are present in the MBFC database (55% if we exclude PolitiFact), meaning the database provides reasonable if not perfect coverage. We also note that it covers all of the top 10 sources discussed above, except for Wikipedia.

On average, the leaning of the sources used is -0.54. This is between center and center-left. We note that PolitiFact is rated at -1.

We pair this with an analysis of the leaning of the input statements, using evaluation by GPT-4. GPT-4 has been shown to effectively measure ideology (Törnberg, 2023; Yu et al., 2023) and source overall domain quality (Yang & Menczer, 2023), with high alignment to MBFC. We prompt it as follows:

```
Given the following statement, analyze and determine the
    political ideology the information would favor if true.
    Note that the information may be false; this should not
    influence your rating, we want to assess the political
    favoring, not the veracity. Consider the use of language,
     themes, and any eplicit or implicit political references
    or endorsements. Provide an estimate of the poltical
    favoring on a scale of -3 to 3, where

-3 indicates an extreme left-wing favoring,
-2 indicates a left favoring,
-1 indicates a center-left wing favoring,
0 inciates center favoring,
1 indicates a center-right wing favoring,
2 indicates a right favoring,
3 indicates an extreme right-wing favoring

Statement: {statement}
Reply with just the score, do not include any additional
    information.
```

This produces an average rating ("GPT Bias Score") of 0.53, which is almost exactly complimentary to the -0.54 of the sources used and might suggest some relationship. We also note the average domain quality for the sources is in the 0.76-0.80 out of 1.00 for all models, indicating that the sources our agent accesses are mostly factual, and highly credible (For reference, New York Times has a rating of 0.87) (Lin et al., 2023). When looking at the distribution of sources for each input leaning, however, we do not find a clear pattern (Figures 6 and 7).

---

[5]https://mediabiasfactcheck.com/
[6]https://rapidapi.com/albertoescobar/api/political-bias-database/

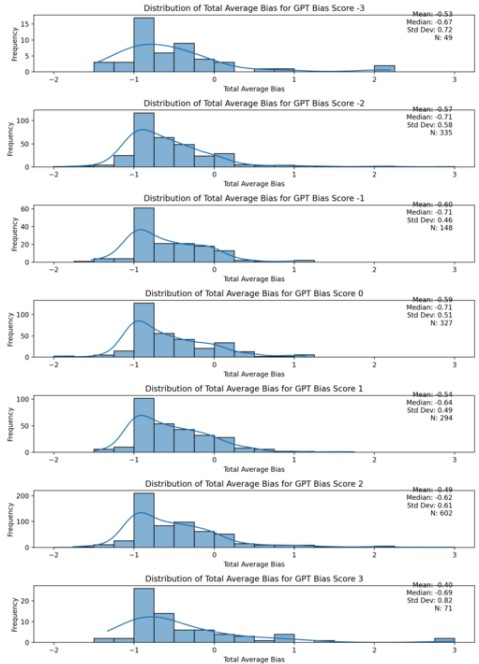

Figure 6: Distribution of source bias, divided by leaning of the input statements as assessed by GPT-4.

We next seek to understand the effects of bias, and overall domain quality of the sources used, and how they relate to bias of the input statement. We again use the MBFC bias ratings for sources. For overall domain quality, we use an aggregate measure of domain quality (Lin et al., 2023) which uses a wisdom of experts approach to evaluate sources, with MBFC being one of multiple sources for domain quality ratings. We analyze cases where the web retrieval agent outperforms the offline model (i.e., the agent without web search capabilities) and cases where web retrieval yields less accurate results compared to the offline model. In particular, we consider the offline model's evaluation to be accurate if the majority of its 5 evaluation trials on a given statement are accurate. Note that we use the same standard for the online (search-enabled) model. We look at cases where the online model is accurate and the offline one is not (i.e., where search is improving performance) and vice versa (where search is worsening performance).

Figures 8 and 9 show the normalized distributions of total average bias and domain quality respectively for GPT-3.5. The green bars represent cases where web retrieval improves performance, while the red bars represent cases where web retrieval worsens performance compared to the offline model. The results are divided by the leaning of the input, assessed by GPT-4 as described above. Similarly, Figures 10 and 11 show the same distributions for GPT-4.

We hoped this analysis might point to a way to improve performance or show other clear patterns, but unfortunately, these results are not conclusive.

In summary, these additional analyses reveal that the sources used are slightly left-leaning, while the input statements—if they were true—are slightly right-leaning. Thus, our system does not appear to have a strong bias in this dimension. We believe these results are valuable for evaluating and comparing different evidence retrieval systems. On the other hand, we did not find clear patterns in bias, credibility, or factuality of sources and inputs that might point to ways to improve performance.

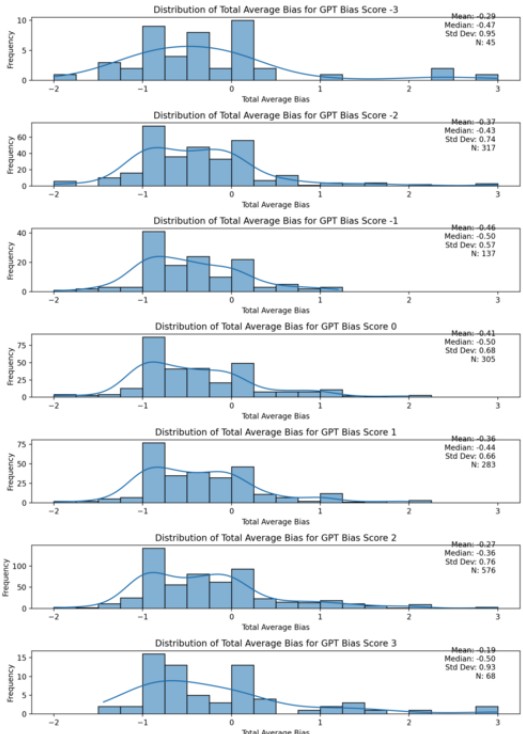

Figure 7: Distribution of source bias, divided by leaning of the input statements as assessed by GPT-4, with PolitiFact excluded.

### A.6 Reliability diagrams

We illustrate the effect of enabling web-search on the system's calibration performance through reliability diagrams. Comparing figures 12a and 12b, and 13a and 13b, we observe how enabling web-search increases our system's uncertainty scores, in parallel with its improvement on calibration.

### A.7 Confidence in search results vs accuracy

We also attempted to perform confidence measures on the search results themselves, and analyze the correlation between those scores versus correctness of the prediction. The hypothesis is that from search results that the model is more confident about, we will get a higher accuracy. Conversely, if the model is less confident about a search query's results, it will have a lower chance of correctly evaluating the veracity of the statement for which the search was conducted. To test this, we use the dataset of duckduckgo search result summaries (Duck –PF GPT 4). We prompt the gpt-4-0125-preview model (with a default temperature of 1) to get confidence scores on a search summary given the original statement and search query, to assess if the result of the search query made was accurate and comprehensive.

We attempted a few prompts. These were put as the system prompts, with the statement being the user prompt.

Prompt 1:

```
On a scale of 0-100, output your confidence that the information
    found in the overall statement is accurate and comprehensive.
    Only give the output.
```

Prompt 2:

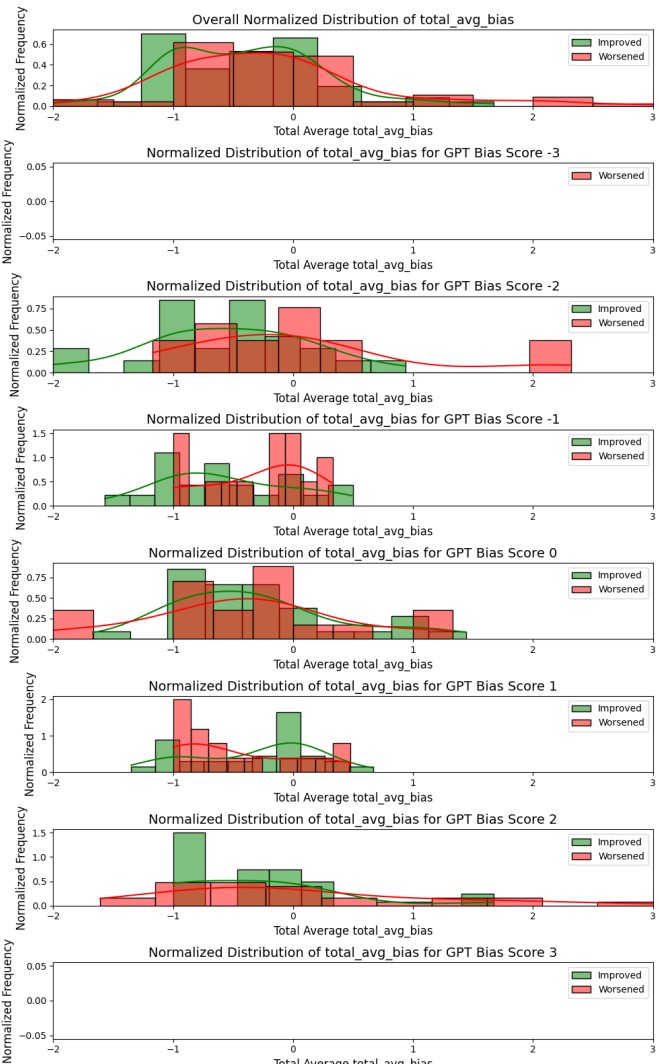

Figure 8: Normalized distribution of total average bias for GPT-3.5 with varying GPT bias scores.

```
On a scale of 0-100, where 0 is highly uncertain and 100 is fully
    certain, output your confidence that the information found in
    the overall result to the search query is accurate and
    comprehensive. Only give the output.
```

Based on negative results from the above attempts, we tried another prompt with additional context, this time mentioning both the original statement and search query along with the search summary, and emphasizing the role the model should play (reflecting the confidence that the search summary matches the search query).

Prompt 3:

```
Based on the following information, output your confidence, on a
    scale of 0-100 where 0 is highly uncertain and 100 is fully
    certain,  that the result of the search query made was
    accurate and comprehensive with regard to the topic queried.
    Don't write an explanation, only give the number.
```

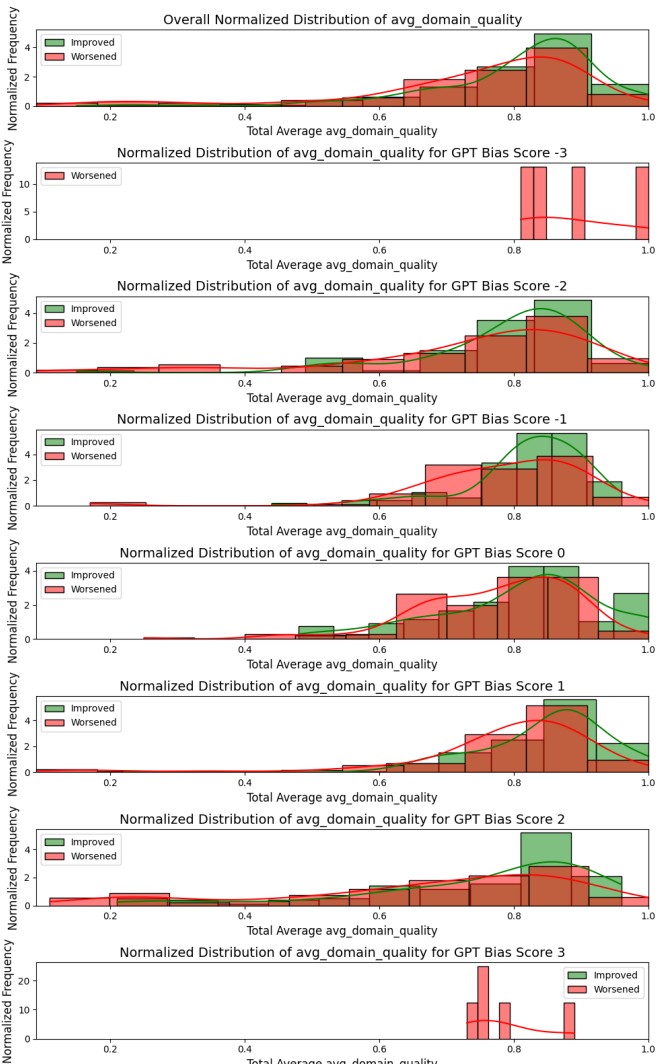

Figure 9: Normalized distribution of average domain quality for GPT-3.5 with varying GPT bias scores.

```
The search query was made as one part of a process to evaluate the
    veracity of this statement: {statement}
{query}
The analysis of the search results by the searching agent was: {
    result_summary}

Note that your score should reflect confidence in the accuracy and
    comprehensiveness with regard to the topic of the search
    query. In other words, the extent to which it found the
    information it sought. It should not reflect whether it
    comprehensively answered everything about the statement,
    because there can be other search queries and other analysis
    taking place too.
```

With all these prompts, the model outputs a confidence score between 0-100, which can be correlated to correctness of the prediction obtained in the final evaluation for each of those search results. The violin plot for Prompt #3 (Figure 14) obtained shows no clear difference

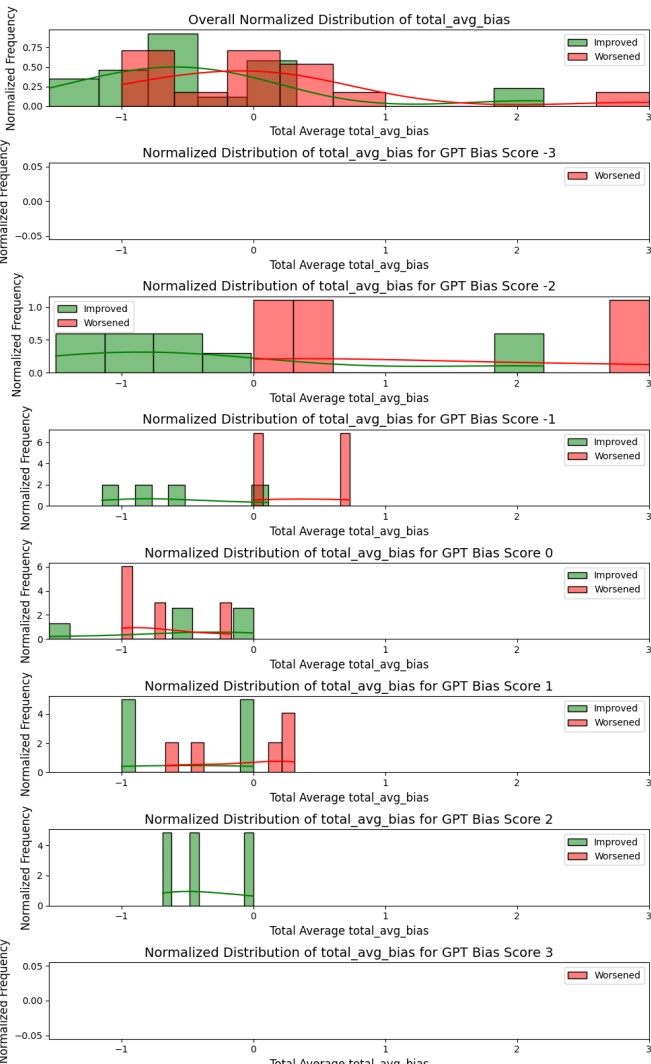

Figure 10: Normalized distribution of total average bias for GPT-4.0 with varying GPT bias scores.

between confidence scores on accurate and inaccurate predictions, and the data had a very low correlation score of -0.02. Thus, these results did not provide effective quantification of the quality or comprehensiveness of search results. We pose such uncertainty quantification as an open problem.

## A.8 Sensitivity to variations in prompt template

LLMs can sometimes be sensitive to variations in prompt templates and formatting changes. To test whether our pipeline is robust and whether the reported metrics provide solid conclusions irrespective of the exact prompt template, we repeated the experiments across five different paraphrased versions of the task instruction. In table 12, we report the performance of both Cohere Search and No Search on a randomly selected 50-example subset of the full LIAR-New dataset. We find that while there can indeed be some variation between different prompts, our search setup successfully improves performance in every case.

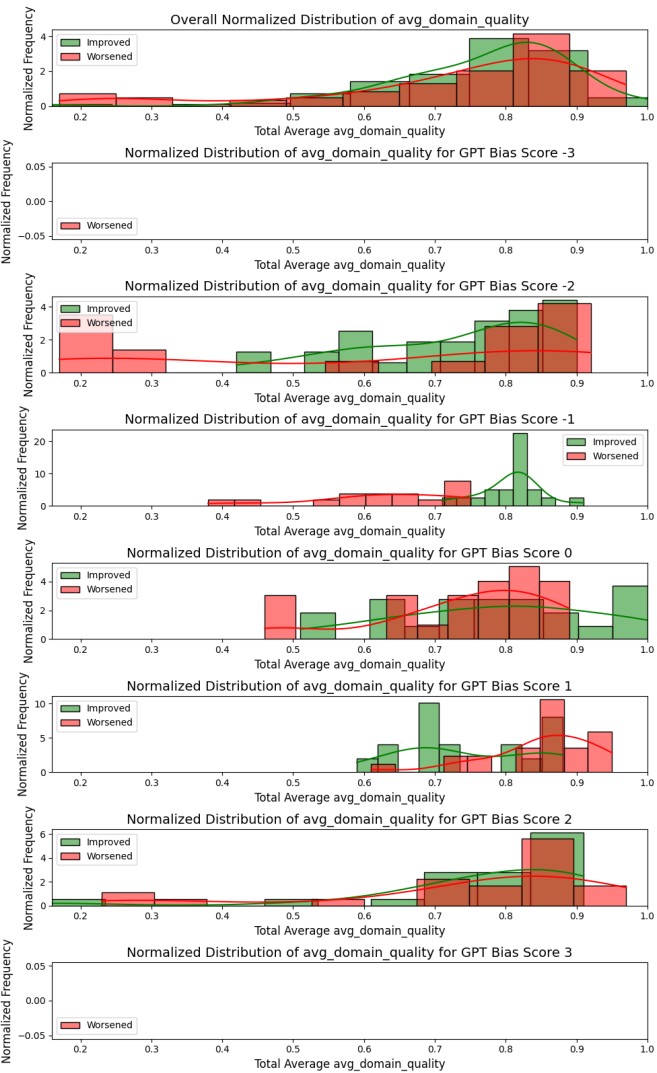

Figure 11: Normalized distribution of average domain quality for GPT-4.0 with varying GPT bias scores.

### A.9 Search tool usage statistics

In our pipeline, we provide the LLM the option to invoke search as many times as needed (up to ten times per query). We also allow the LLM to not execute any query at all. In table 13, we report the average number of search requests per query on LIAR-New, grouped by whether the model produced the correct prediction. We see slightly more queries on examples that ended in incorrect predictions, which could suggest those examples are more complex or otherwise more difficult to retrieve the right information.

### A.10 Results on datasets within training knowledge cutoff

Unlike many other fact-checking datasets, LIAR-New is unique in that it contains exclusively fact-checks collected outside the training knowledge cutoff of GPT-3.5 and the older GPT-4 LLM (gpt-4-0613). Performance on such datasets provide hints at the effectiveness of our pipeline in verifying novel claims if a system like ours is deployed in the real world.

In Table 14, we report results on the older datasets FaVIQ, X-FACT, FEVER-v1, and the validation set of FEVER-v2. In all these datasets, search appears to yield a performance

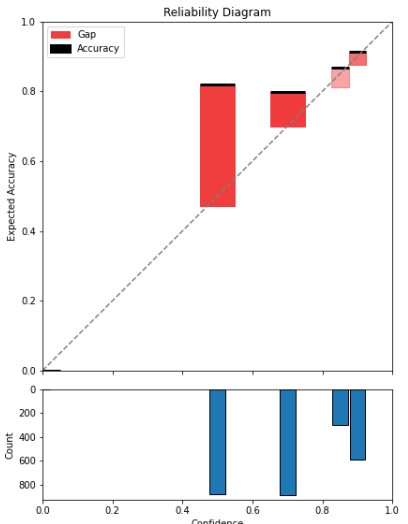

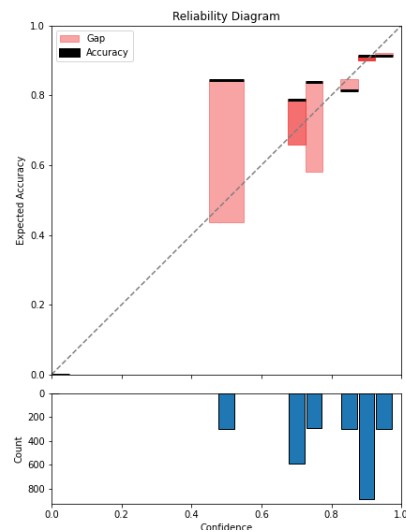

(a) Reliability diagram for gpt-3.5-turbo without search on LIAR-New.

(b) Reliability diagram for gpt-3.5-turbo with search on LIAR-New.

Figure 12: Comparison of reliability diagrams for gpt-3.5-turbo with and without search on LIAR-New.

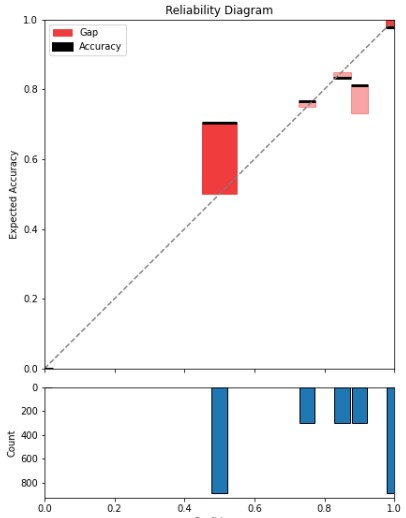

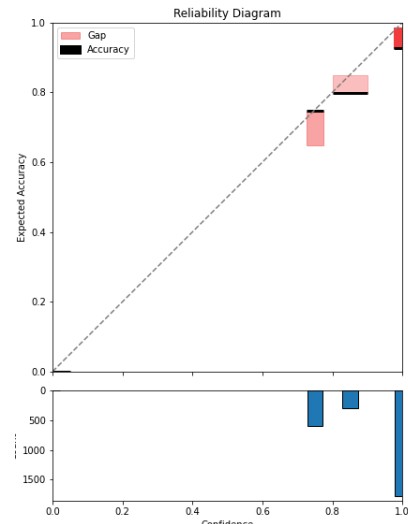

(a) Reliability diagram for gpt-4-0125 without search on LIAR-New.

(b) Reliability diagram for gpt-4-0125 with search on LIAR-New.

Figure 13: Comparison of reliability diagrams for gpt-4-0125 with and without search on LIAR-New.

improvement but only a very marginal one. This is nonetheless a positive result because retrieving evidence without compromising performance provides its own benefit, since it enables citing sources and providing users followup materials. We hypothesize that the performance benefit is low because many questions in these datasets appear too easy to the model, either due to the knowledge cutoff and perhaps appearing in the training data, or just the overall perceived difficulty of the questions themselves. For instance, we observed the model explicitly saying search was not necessary on some examples, and the rates of

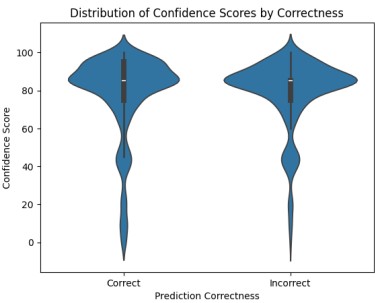

Figure 14: Violin plot of confidence scores over accuracy (Prompt #3).

| Paraphrased Instruction | No Search | Cohere Search |
|---|---|---|
| Your task is to analyze the factuality of the given statement. Once you have detailed your analysis process, conclude with a summary. If you determine the statement is accurate, state "True statement; Factuality: 1." If you find the statement to be inaccurate, state "False statement; Factuality: 0." Start your summary with the word "Summary: ". | 50.0% ± 17.7% | 63.3% ± 25.3% |
| Your job is to evaluate the **accuracy of the provided** statement. \n\nOnce you have detailed your analysis process, conclude with a summary. If you determine the statement is accurate, state "**Summary: True statement; Factuality: 1**". If **the statement is inaccurate**, state **"Summary: False statement; Factuality: 0**". | 57.3% ± 13.4% | 69.6% ± 28.5% |
| Your job is to evaluate the **accuracy of the provided** statement. \n\nOnce you have detailed your analysis process, conclude with a summary. If you determine the statement is accurate, state "True statement; Factuality: 1." If **it is not accurate**, state "False statement; Factuality: 0." Start your summary with the phrase "Summary: ". | 52.9% ± 15.5% | 68.1% ± 15.2% |
| Your job is to evaluate the **accuracy of the provided** statement. \n\nOnce you have detailed all the steps of your analysis, conclude with a summary. If you determine the statement is accurate, state "True statement; Factuality: 1." If you find the statement to be inaccurate, state "False statement; Factuality: 0." Start your summary with the word "Summary: ". | 62.8% ± 8.8% | 61.9% ± 18.5% |
| Your job is to evaluate the **accuracy of the provided** statement. \n\nOnce you have detailed all the steps of your analysis, conclude with a summary. If you determine the statement is accurate, state "True statement; Factuality: 1." If you find the statement to be inaccurate, state "False statement; Factuality: 0." Start your summary with the word "Summary: ". | 53.1% ± 12.7% | 59.1% ± 15.4% |

Table 12: Instruction paraphrasing impact on performance.

search are low compared to LIAR-New. Thus, the current system seems to yield a consistent benefit, but the degree - and whether it is in performance or primarily adding evidence - varies depending on the dataset. In future work we plan to investigate if pushing the system to search more on these datasets would yield a larger increase in performance.

| Model | Liar-New | |
| | Incorrect | Correct |
|---|---|---|
| vicuna | 1.2 | 1.1 |
| mixtral | 1.7 | 1.4 |
| haiku | 1.6 | 1.6 |
| gpt-3.5 | 1.4 | 1.2 |
| gpt-4 | 1.2 | 1.1 |

Table 13: Average number of model requests to Cohere-RAG search per query for Liar-New.

| Dataset | No Search F1 | Search via Cohere RAG | | | ΔF1 |
| | | F1 | Incorrect/Search | Correct/Search | |
|---|---|---|---|---|---|
| faviq | 77.7% ± 1.9% | 78.0% ± 2.6% | 0.54 per example | 0.61 per example | +0.3% |
| x-fact | 56.8% ± 1.9% | 57.3% ± 1.6% | 0.43 per example | 0.45 per example | +0.5% |
| fever | 93.3% ± 1.0% | 93.7% ± 1.4% | 0.64 per example | 0.38 per example | +0.4% |
| fever-v2-validation | 84.6% ± 0.0% | 84.7% ± 0.2% | 0.90 per example | 0.91 per example | +0.1% |

Table 14: F1 score of gpt-4-0125 (2023/12) with and without search on FaVIQ and X-Fact.

## A.11 Example of multi-step claim decomposition

As seen in the prompt template, we did not explicitly prompt the main agent LLM to decompose the input statement that needs to be verified. Nevertheless, when prompted to provide analysis steps and to make use of a search engine to look up relevant sources, we observed that in quite a few instances, the LLM decomposed the input statement into a number of sub-claims. As seen in the following example, the sub-queries are typically more self-contained and hence potentially easier to look up using the search tool than the original statement. We hypothesize that this capability might explain at least in part why main agents that are more powerful seem to yield greater performance gain from the use of the search tool.

In general, we observe that the capability of the generator model might create a bottleneck for the overall performance of the pipeline. For instance, while both gpt-3.5-turbo and gpt-4-0613 had the same knowledge cutoff, gpt-4-0613 demonstrated a significantly better performance gain. A preliminary review of the model generations suggested that queries from gpt-4-0613 were more concise, allowing it to retrieve more relevant results from web searches. While additional analyses are required, we note that the quality of the generated search queries might be the reason why adding search to vicuna-13b-v1.5 did not improve overall performance by much.

Additionally, we note that the performance gain from search is more pronounced with gpt-4-turbo-0125 (+12.8% F1, 2023/12 cutoff) as the generator model than with gpt-3.5-turbo (+5.4% F1, 2021/03 cutoff). We find this result surprising, especially given that when compared with gpt-3.5-turbo, gpt-4-turbo-0125 has a more recent knowledge cutoff. Specifically, the LIAR-New dataset includes only statements from fact-checking articles from October 2021 and later, meaning the world knowledge of gpt-4-turbo-0125 (up to 2023/12) encompasses the dataset's entire time span, while that of gpt-3.5-turbo (up to 2021/03) includes none of it. Our initial hypothesis was that this model would benefit less from web search, but that was not the case. We suspect that this unexpected trend might be related to the relative differences between the logic capabilities of these two LLMs, as well as the limited capacity of the models to comprehensively memorize and retrieve facts from training. However, additional work will be required to fully understand this difference.

- **User:** Your task is to analyze the factuality of the given statement. You have access to a search engine tool. To invoke search, begin your query with the phrase "SEARCH: ". You may invoke the search tool as many times as needed.

  After providing all your analysis steps, summarize your analysis and state "True statement; Factuality: 1" if you think the statement is factual, or "False statement; Factuality: 0" otherwise. You should begin your summary with the phrase "Summary: ".

  Statement: Comparing the price of oil and gas in June 2008 to March 2022 shows that oil companies are price gouging.

- **Assistant:** To analyze the factuality of the statement, I will first need to gather data on the prices of oil and gas in June 2008 and March 2022. Then, I will need to understand the definition of "price gouging" to see if the pricing behavior by oil companies during these periods qualifies.

  SEARCH: price of oil June 2008

- **Cohere Search Connector:** Search result: I found multiple prices for oil in June 2008. One source stated that oil prices fell from a high of $133.88 per barrel in June 2008 to a low of $39.09 per barrel in February 2009.

  However, another source stated that on June 30th, 2008, the price of Brent Crude was $132.32 per barrel, but the price on June 27th was $140.21 per barrel, while the price was $122.8 dollars per barrel in May 2008. These figures represent the price of oil per barrel, and not the price per gallon. Unfortunately, I was unable to find information regarding the price of oil in June 2008 in USD/gallon.

- **Assistant:** Given the search result, we have a general understanding of the price of oil in June 2008, which fluctuated around $132 to $140 per barrel. For a comprehensive analysis, we also need the prices in March 2022.

  SEARCH: price of oil March 2022

- **Cohere Search Connector:** Search result: Oil prices soared to heights not seen since 2008 due to the Russia-Ukraine war, with ICE Brent oil futures sliding to around $100/bbl intraday.

  Other sources:

  - US Energy Information Administration, Short-Term Energy Outlook, March 2022: Brent crude oil is expected to be $116 per barrel for the second quarter of 2022.

  - Daily Oil Price: March 31, 2022, Crude Oil: 100.28 (-7.54).

- **Assistant:** With the provided information, we can compare the price of oil in June 2008 and March 2022. In June 2008, the price of oil per barrel was around $132 to $140, and in March 2022, it ranged from around $100 to $116 per barrel according to different sources. The prices in both periods were high, partially due to geopolitical tensions (2008 and 2022 respectively).

  To determine if this constitutes "price gouging," we need to understand the term. Price gouging typically refers to when a seller increases the prices of goods, services, or commodities to a level much higher than is considered reasonable or fair, often during a demand surge or supply constraint. It's crucial to note that the concept of price gouging is often related to legal and ethical considerations, which may vary by jurisdiction.

  Given the information, the statement connects high oil prices directly to "price gouging" by oil companies. However, determining price gouging involves an analysis of various factors, including production costs, market demand, and geopolitical contexts, among others. The high prices in both 2008 and 2022 are attributed to significant geopolitical events that likely affected global supply and demand, rather than direct evidence of oil companies engaging in price gouging.

  Summary: The statement simplistically attributes the high prices in June 2008 and March 2022 directly to price gouging by oil companies without considering the complexities surrounding global oil prices, including major geopolitical events and supply-demand dynamics. While the prices were indeed high in both periods, attributing these solely to price gouging without further evidence oversimplifies the issue. False statement; Factuality: 0.

### A.12 LIAR-New F1 for each label class

In Table 15, we present a per-class view into how the addition of Cohere Search affects the F1 scores on Liar-New, depending on whether the given statement is true or false. The information below is produced using scikit-learn.[7]

| Main Model (Knowledge cutoff) | No Search | | Cohere Search | |
|---|---|---|---|---|
| | False | True | False | True |
| vicuna-13b-v1.5 | 87.86 | 32.40 | 88.80 (+0.94) | 34.48 (+2.08) |
| mixtral-8x7b-instruct-v0.1 | 91.11 | 11.21 | 91.18 (+0.07) | 14.55 (+3.34) |
| command search connector | 90.14 | 38.27 | 91.92 (+1.78) | 53.18 (+14.91) |
| command no connector | 81.33 | 39.13 | 85.63 (+4.30) | 30.69 (-8.44) |
| gpt-3.5-turbo (2021/03) | 91.55 | 27.64 | 92.13 (+0.58) | 44.90 (+17.26) |
| gpt-4-0613 (2021/03) | 93.05 | 0.00 | 93.10 (+0.05) | 53.85 (+53.85) |
| gpt-4-0125-preview (2023/12) | 91.60 | 23.93 | 92.76 (+1.16) | 51.95 (+28.02) |
| claude-3-haiku (2023/08) | 90.10 | 38.79 | 91.54 (+1.44) | 50.29 (+11.50) |

Table 15: Per-class F1 scores (scaled by 100) for different models with and without search, with differences relative to "No Search". gpt-4-0613 (2021/03) is evaluated on a subsampled set of 100 examples (89 False, 11 True). All other models are evaluated on a randomly selected subset of 588 examples (495 False, 93 True, 30% of liar-new), which encompasses the 100 examples mentioned above.

---

[7]https://scikit-learn.org/stable/modules/generated/sklearn.metrics.classification_report.html

