# OpenReview forum: "Web Retrieval Agents for Evidence-Based Misinformation Detection"
_colmweb.org/COLM/2024/Conference — COLM_

### Official Review · Reviewer_SMvV · 2024-05-06

**Rating:** 5
**Confidence:** 4
**Ethics Flag:** 1

**Summary:**

This paper proposes an approach that combines web retrieval with LLMs for misinformation detection. The paper presents a range of experiments that varies different components of the pipeline. Overall this is an important direction of research, and the results are promising. However, I had trouble reading this paper as it does have severe presentational issues, partly due to the many analyses; they often lacked explanation, depth or certain choices were not clearly explained. Some models appeared in one analysis, but not in the other. A table is shown but barely explained (Table 7). Furthermore, some natural questions could have been pursued (see my comments) to shed further light on this topic, such as an in-depth analysis of the type of queries that the LLMs execute.

**Questions To Authors:**

Not necessarily questions, but some remaining comments:

* What do you mean with HiSS being unstable?
* Page 6 is hard to read due to the many tables (caption vs. text is not clear)
* I wish the authors had done an analysis on the queries executed; this would really strengthen their argument about decomposition and highlight future directions of research. What queries are executed by the LLMs? Are they relevant? (basically: do they target the information that is needed?). Currently, it is also unclear where the errors come from. Are the queries not suitable, or does the LLM summarization part go wrong?
* There are certain design changes that are not clearly explained. As an example, why use GPT-3.5 to summarize the prompt? Wouldn’t a more natural choice be to have the summarization model match the main LLM?

Some details of the experiments are unclear:
* You experiment with different summarization models in Table 5, but it’s unclear which LLM is used as the main model there.
* The PolitiFact experiment is nice, but why did you only report results with GPT-3.5-turbo as the main model, while that’s a model where the improvement using search was already low (so then perhaps the effect of removing PolitiFact is also limited?). A more natural choice would be to do this for all models, or for the best one (gpt-4).
* Causal Walk (Zhang et al. (2024)) is used as a baseline in Table 7, but this was never explained/introduced before.
* The difference between the proposed approach and other baselines (e.g., HiSS, WikiChat etc.) can be explained more clearly.

**Reasons To Accept:**

* Fact-checking is a very important and timely topic. Furthermore, (as the paper claims), retrieval-augmented LLM approaches have been barely explored so far, while they make perfect sense for this problem.
* The paper includes a wide range of experiments (but this is also a weakness); I particularly liked the missing information analysis.
* The paper could inspire future work.

**Reasons To Reject:**

* Presentational issues: This in itself is not a reason to reject the paper, but it did weaken the paper. As an example, the references miss parentheses throughout the text, which makes reading the paper more difficult. The paper also suddenly includes results with the Fever v2 dataset, but this was never introduced. See more comments below.
* It is unclear how the prompts were developed. Furthermore, studies have shown that results can be sensitive to only small changes to the prompts (even small formatting changes); it’s therefore not clear to me how much the results would change with small prompt changes.
* The main experiments use only one dataset (LIAR-New), and actually only a subset of this dataset. Given that misinformation is such a large topic with many datasets available, I would expect experiments on more datasets. Otherwise, the authors need to clearly explain why the other datasets are not suitable. Even if it’s the case that this dataset offers a unique benefit (e.g., labels for examples with missing context), other parts of the experiments can be performed on other datasets as well to strengthen the analysis.
* I believe the paper would have been stronger if it would only focus on a few core analyses, rather than trying to include so many as it does now. In many cases, the analyses lack explanation or depth, or are not systematic enough (e.g., only performed on a subset of the models, without explaining why).

---

> ### Author Rebuttal · Authors · 2024-05-31
>
> Thank you for the constructive criticism. We address some of your key points below; we will respond to others during the discussion period when we are not constrained by character limit.
>
> 1. The issues you pointed out in the writing and completeness make sense. We have made and will continue to make improvements, including but not limited to: fixing the citations, adding introduction of Fever v2 and the Zhang et al. (2024) method we compare with; additional explanations of the differences between our proposed approach and others; and the model used for the main agent in Table 5 (GPT-3.5).
> 2. Regarding prompt sensitivity: we rephrased the instructions in five different formats using gpt-4o, for both search and no-search. We observed consistent performance gain in the search experiment over no search. The no-search experiments showed greater variability in F1 and majority vote (5-fold) F1 (0.451 to 0.642) vs the search-enabled experiments (0.621 to 0.751), suggesting that the additional context from search might have reduced the sensitivity to the exact prompt that was used.
> 3. LIAR-New has advantages such as being beyond old GPT-4’s main knowledge cutoff and Possibility labels, but the datasets tested were largely restricted by budget (like some of the other analyses, such as running all experiments with all models). We have now run preliminary experiments on FaVIQ and X-Fact with GPT-4 and Cohere search. In both cases search improves performance (76.6% ± 4.4% vs. 77.5% ± 8.3%, 57.9% ± 8.7% vs. 63.1% ± 21.3%, respectively). We will report on more finalized results with larger sample size soon in the discussion period, and will add these experiments to the paper.
> 4. Besides the significant limitation of worse results with a better model (GPT-4 vs. GPT-3.5), we consider HiSS unstable because it does not output a proper answer in ~9% of cases in both the original version and GPT-4 version (Table 9). This is a much larger failure rate than ours, with equivalent models falling well under 2%. We also note that the original version of the HiSS code had an even higher failure rate. After bringing this to the authors’ attention, they released an updated version, which still exhibited the significant failure rate reported in Table 9.
>
> Thank you for your consideration and we would welcome further discussion.

---

> > ### Comment · Reviewer_SMvV · 2024-05-31
> >
> > Thanks for the response! I've increased my rating to 5.
> >
> > As a minor comment: It's still not clear to me what you mean with 'unstable' (what is your definition of it?)

---

> > ### Author Response · Authors · 2024-06-05
> >
> > Thank you for considering our response and revisiting your rating!
> >
> > We said "unstable" in the sense that it does not give stable performance across variations that one would not expect or desire to damage performance (most notably, using a significantly stronger LLM). Both you and Reviewer NQTb found this unclear, however, so it is probably a poor choice of words. We will revise it to say more directly "its performance is mixed, having worse results with GPT-4 than GPT-3.5, and a high rate of not outputting an answer as requested."
> >
> > Here are some additional results and responses to more of your original points:
> >
> > 1. We've run Claude 3 Haiku as main agent and found it gives promising results, achieving 71.3% ± 6.6% F1 on LIAR-New with Cohere search. This is roughly equal to GPT-4 as main agent.
> >
> > 2. We have done analysis on how many search queries and reasoning steps (not including search results) on average that the LLMs go through, and how they relate to correct answers. Some of the key results on LIAR-New include:
> >
> > | model | Incorrect Reasoning | Incorrect Search | Correct Reasoning | Correct Search |
> > | -------- | ------- | -------- | ------- | -------- |
> > | vicuna |	4.3| 	1.2| 	4.2| 	1.1|
> > | mixtral |	5.4	| 1.7	| 4.8	| 1.4|
> > | gpt-3.5 |	4.7 | 1.4 |	4.5	| 1.2 |
> > | gpt-4 |	3.9| 	1.2	| 3.8	| 1.1|
> > | haiku |	4.2	| 1.6	| 4.3 |  1.6 |
> >
> > GPT-4 does the least reasoning, on par with the least search, and the least number of total tokens. This might suggest it is more efficient and can rely more on its superior knowledge base and reasoning. Conversely, Vicuna, with approximately tied least search and second lowest reasoning steps per example, may be providing poor performance because it does not take enough advantage of search to compensate for its insufficient reasoning or knowledge.
> >
> > Mixtral and Haiku clock in with the most search and reasoning steps. GPT-3.5 falls in the middle. Potentially, Haiku is able to achieve performance near GPT-4 by combining both a stronger model than GPT-3.5 and Mixtral, and above average amounts of search and reasoning.
> >
> > Across nearly all models, more search and more reasoning steps are used for incorrect predictions than correct ones. This is a bit counterintuitive and could indicate either that these examples are the most complicated, or are ones where the models are struggling to find the right information and searching repeatedly. Either case could point to a way to improve results in future work.
> >
> > We also plan to do an experiment replacing the final predictive step of Vicuna with GPT-4 and vice-versa, thus feeding each model the search queries and intermediate reasoning of the other, which should help answer which part of the process is key to performance. Unfortunately, this result will probably not be completed in time for the discussion period.
> >
> > 3. The point about the plus and minus PF results being run on GPT-3.5, where the impact of search was less pronounced, makes sense. As noted in response to another review but missing in original response to yours, we've now run Duck -PF with GPT-4. In this case, we do find performance decreases somewhat, from 70.3% ± 8.5% to 66.9% ± 6.5% with k=10. However, it remains much better than no search. With k=5 it decreases to 64.7 ± 4.3, and 63.7 ± 3.4 with k=2, following the trend observed with GPT-3.5.
> >
> > As before, thank you for your time and we'd be happy to discuss if you have further thoughts.

---

> > > ### Author Response · Authors · 2024-06-06
> > >
> > > Update on results on other datasets: with a larger sample size, the results of GPT-4 are now 78.0% ± 2.6% vs. 77.7% ± 1.9% respectively with and without search on FaVIQ, and 57.3% ± 1.6% vs. 56.8% ± 1.9% with and without search on X-Fact. Thus, search appears to yield a performance improvement but only slight on these datasets. This is nonetheless a positive result because retrieving evidence with greater or equal performance provides its own benefit, since it enables citing sources and providing users followup materials. We hypothesize that the performance benefit is low because many questions in these datasets appear too easy to the model. Specifically, we observed the model explicitly saying search was not necessary on some examples, and the rates of search are very low compared to LIAR-New: 0.54 and 0.61 average queries for FaVIQ examples it got correct and wrong, and 0.45 and 0.43 on X-Fact, contrasted with more than 1 query per example average on LIAR-New. Thus, the current system seems to yield a consistent benefit, but the degree - and whether it is in performance or primarily adding evidence - varies depending on the dataset. In future work we plan to investigate if pushing the system to search more on these datasets would yield a larger increase in performance.

---

### Official Review · Reviewer_ZeNQ · 2024-05-09

**Rating:** 7
**Confidence:** 4
**Ethics Flag:** 1

**Summary:**

This paper proposes a framework for misinformation detection based on agentic AI and LLMs, where LLMs are enabled to decide when to search for additional information and retrieve it through web search in an iterative process. The paper studies the impact of such framework over the LIAR dataset. It also studies the effect of the retrieval and summarization tools used as well as the relevance of the sources where additional information is retrieved from.

**Reasons To Accept:**

- Innovative approach combining agentic AI and RAG
- Thorough analysis from different perspectives although I miss that the evaluation also addressed other datasets in addition to LIAR

**Reasons To Reject:**

- Evaluating on more evaluation datasets would have provided a wider perspective on the approach and its possible generalization

---

> ### Author Rebuttal · Authors · 2024-05-31
>
> Thank you for the positive feedback, and raising the point about adding more datasets for a wider perspective and better understanding of generalization. The datasets tested were largely restricted by budget, but your point is valid nonetheless. We have now run preliminary experiments on FaVIQ and X-Fact with GPT-4 and Cohere search. In both cases search improves performance (76.6% ± 4.4% vs. 77.5% ± 8.3%, 57.9% ± 8.7% vs. 63.1% ± 21.3%, respectively). We will report on more finalized results with larger sample sizes soon in the discussion period, and will add these experiments to the paper. If you have any further thoughts here or on the paper in general we’d be happy to discuss.

---

> > ### Comment · Reviewer_ZeNQ · 2024-06-04
> >
> > Thank you for your response. I think this will contribute to increase the robustness of your findings.

---

> > > ### Author Response · Authors · 2024-06-06
> > >
> > > With a larger sample size, the results of GPT-4 are now 78.0% ± 2.6% vs. 77.7% ± 1.9% respectively with and without search on FaVIQ, and 57.3% ± 1.6% vs. 56.8% ± 1.9% with and without search on X-Fact. Thus, search appears to yield a performance improvement but only slight on these datasets. This is nonetheless a positive result because retrieving evidence with greater or equal performance provides its own benefit. We hypothesize that the performance benefit is low because many questions in these datasets appear too easy to the model. Specifically, we observed the model explicitly saying search was not necessary on some examples, and the rates of search are very low compared to LIAR-New: 0.54 and 0.61 average queries for FaVIQ examples it got correct and wrong, and 0.45 and 0.43 on X-Fact, contrasted with more than 1 query per example average on LIAR-New. Thus, the current system seems to yield a consistent benefit, but the degree - and whether it is in performance or primarily adding evidence - varies depending on the dataset. In future work we plan to investigate if pushing the system to search more on these datasets would yield a larger increase in performance.

---

### Official Review · Reviewer_tCVr · 2024-05-10

**Rating:** 7
**Confidence:** 3
**Ethics Flag:** 1

**Summary:**

The paper examines the problem of misinformation detection through experiments that compares and combines various LLM models (e.g. GPT-4) with two RAG search systems (Cohere and DuckDuckGo). The paper evaluates the various system configurations they constructed  primarily on the LIAR-New dataset which some additional experiments using the FEVER dataset.

The paper finds that the best approach involves using a SOTA LLM like GTP-4 with prompted instructions to craft queries to submit to the RAG search systems and then analyzing the summarized RAG search results against the original information claim to make a final decision.

Several interesting observations were made along the way, including one which indicate than open web searches do not see degradations in finding accurate/truthful information on the evaluation data sets even when reliable information sources like Politifact and Wikipedia are disabled in the search.

**Reasons To Accept:**

Very interesting paper. I enjoyed reading it.

Solid experiments comparing numerous different system configurations and different forms of ablation.

**Reasons To Reject:**

I don't see any reason to reject. The approach taken is simple, yet effective and the experiments provide multiple comparative studies.

---

> ### Author Rebuttal · Authors · 2024-05-31
>
> Thank you for reviewing our paper, and appreciate the positive feedback!

---

### Official Review · Reviewer_NQTb · 2024-05-11

**Rating:** 6
**Confidence:** 4
**Ethics Flag:** 1

**Summary:**

The paper presents a system that combined retrieval-augmented generation (RAG) with large language models (LLMs) for misinformation detection. The authors present a considerable amount of discussion and analysis, reporting significantly improved results on the LIAR-New corpus. The paper presents experiments supporting the flexibility of the presented pipeline, which includes search and summarization modules.

**Questions To Authors:**

1. Please use forward referencing in the introduction to point out where a contribution is detailed. Not just in the listed contributions, but also in the second-half of the first page (just before the contributions are listed).

2. Please clearly define what the plus/minus margins in the tables indicate. Are these standard errors?

3. In Table 4, I was wondering why there are no Duck + PF ablation results for k = 5 and k = 2. Could you please elaborate on this?

4. The font size in Fig. 1 is way too small. Please see if it's possible to increase this for readability.

**Reasons To Accept:**

1. The overall contribution of RAG+LLM with the application being misinformation detection, is a rather novel contribution as an application.

2. Strong analysis and discussion (e.g., the analysis on the type of missing information is certainly a strength) and ablation studies (on the inclusion of Politifact and on the number of search results from DuckDuckGo). In particular, I liked the analysis of the summarizer's role in the pipeline.

3. A good discussion of limitations by models.

4. The inclusion of the analysis of bias and credibility of sources is a definite strength.

**Reasons To Reject:**

There are a few drawbacks, which I list and discuss here. Minor remarks and suggestions are provided separately as "questions to authors":

1. There are two or three places (Sec. 2, 4, and Appendix A.2) where the authors mention the instability of HiSS. However, I am not sure if there is enough evidence of instability demonstrated in the experimental results. The only indication provided is that the performance of HiSS is worse with GPT-4 than with GPT-3.5. But why is this "unstable"? On a related note, I don't understand why in the baseline models' results in Table 2, there are no errors/deviations reported (except for Cohere Chat with RAG).

2. Some of the auxiliary contributions do not seem novel. For example, suggesting that "no single source is necessary for accuracy, but it is important to have enough sources to reduce noise and improve the chances of finding the necessary evidence." (Sec. 4, "The Role of the Sources"). This observation has been made quite consistently since early work on credibility and truth discovery. On the other hand, this opens up questions about how to reconcile potentially conflicting evidence arriving from different sources.

3. It is not clear if the facts, as discussed in Sec. 3.2 are "atomic" claims or complex claims that be reduced to sub-claims. If they are not atomic claims, then the use of 0/1 binary labels is problematic. In the introduction, it is mentioned that the pipeline integrated techniques for claim decomposition, but I do not see any further details of this in the rest of the paper.

4. In 3.4, the use of macro-averaged F1 is justified. But why not report classwise precision and recall? That will give the readers a more detailed picture of the models' performances.

5. I would also like to point out that the use of Media Bias Fact Check (MBFC) as the only source for bias analysis is questionable. While this resource has been successfully used in NLP literature quite often, it has received severe criticism elsewhere, including journalism. Even Poynter Institute (the institute behind PolitiFact, which plays a key role in this paper) has criticized MBFC's methodology "is in no way scientific." (Funke and Mantzarlis, 2018).

---

> ### Author Rebuttal · Authors · 2024-05-31
>
> Thank you for the feedback and thoughtful comments. We have made some changes to the writing and completed some new experiments to improve the quality of our original submission. Because of character limit in rebuttal, we include some of the most significant ones here, and will share additional responses in the discussion.
>
> 1. Regarding HiSS, worse results with a better model (GPT-4 vs. GPT-3.5) is a big limitation, because it suggests this method will not scale with future LLMs, whereas our method will. In addition, we consider this approach unstable because it does not output a proper answer in ~9% of cases in both the original version and GPT-4 version (Table 9). This is a much larger failure rate than ours, with equivalent models falling well under 2%. We also note that the original version of the HiSS code had an even higher failure rate. After bringing this to the authors’ attention, they released an updated version, which still exhibited the significant failure rate reported in Table 9. For the lack of error bars, we did not perform additional runs for cost reasons, given performance was already poor.
>
> 5. We reran source quality analysis with the repository of https://academic.oup.com/pnasnexus/article/2/9/pgad286/7258994. This study compared rating sets of news domains and found high correlations, generated a comprehensive aggregate rating for 11,520 domains, and confirmed the reliability of expert consensus in evaluating news quality. We find that with gpt3.5-turbo and gpt4 combined with cohere search, mean quality of domains is high (0.787±0.12, 0.796±0.13 respectively, even when excluding politifact). With respect to the comprehensive scoring evaluation, the average quality of sources accessed is on par with Sky News (0.78) and just below NY Times (0.86). This confirms that this setup is generally finding credible sources.
>
> 7. We did not test Duck +PF for k=5 and k=2 for cost reasons, and because if PF is included in small k, the results may become very sensitive to whether PF is found in the search. However, we’ve now run Duck -PF with GPT-4. In this case, we do find performance decreases somewhat, from 70.3%±8.5% to 66.9%±6.5% with k=10. However, it remains much better than no search. With k=5 it decreases to 64.7±4.3, and 63.7±3.4 with k=2, following the previous trend.
>
> Thank you for your time. We look forward to sharing additional responses, and would love to hear your thoughts on these changes, as well as any other feedback you may have.

---

> > ### Author Response · Authors · 2024-06-05
> > **Additional Updates**
> >
> > Here are some additional updates addressing your other points.
> >
> > 4. In general, if you feel there are any key works our findings reinforce and we should include, we would be happy to add discussion and references. Regarding the point about “no single source” in particular, we felt that it was important to verify that our approach works without searching PolitiFact. In some cases there are existing fact-check articles to draw on, but in other cases there are no such sources, and we sought to develop a method that worked in both scenarios. This type of analysis has been omitted in some literature like HiSS (open web) and the many papers that draw on Wikipedia (fixed and not necessarily comprehensive knowledge base). It is reasonable though that more sources should be helpful up to a point. Also, it may be that the reason using a stronger summarizer performs better (Table 5) is that it is better at reconciling conflicting sources like you highlighted; more broadly, this would be an exciting topic for future work.
> >
> > 5. The claims are generally atomic to the extent that they have verdicts from professional fact-checkers and are commonly evaluated with binary labels (e.g., true/false). In particular, LIAR-New is sourced in nearly the same way as the very widely used - and widely used with binary labels - LIAR dataset; it just includes more recent examples. Likewise, FEVER v2 is also commonly evaluated with binary labels. Therefore, our approach follows standard practices to facilitate comparison between methods.
> > We found that although not explicitly instructed to do it, our method naturally leads the LLM to decompose its analysis through multiple steps of reasoning and multiple searches. For example, the interaction shown in Table 1 is evaluating the statement “Comparing the price of oil and gas in June 2008 to March 2022 shows that oil companies are ‘price gouging.’” We see in Table 1 how it searches for 2008, and then after that (not shown in Table) it searches for 2022. We will add some examples and clarify the writing in the paper.
> >
> > 6. We have now added class-wise precision and recall to the appendix. Please let us know if there are any particular cases you would like to see here.
> >
> > 7. We have adjusted the writing to include forward referencing, better font size in Fig. 1, and description of the plus/minus margins. The latter are 95% confidence intervals over 5 runs.
> >
> > Thank you again for your time, and as before would love to hear your thoughts on these changes or any other feedback you may have!

---

### Decision · Program_Chairs · 2024-07-10

**Decision:**

Accept

**Comment:**

Reviewers generally recognize that RAG+LLM for detecting misinformation is a novel contribution as an application. Questions and comments from the reviewers regarding some phrasing and details of the experiments are addressed properly in a detailed rebuttal. The authors are encouraged to revise the paper in the next version to fully address the reviewers' concerns.